# Scheduling Optimization of Offshore Oil Spill Cleaning Materials Considering Multiple Accident Sites and Multiple Oil Types

**Kai Li** [1,2,3], **Hongliang Yu** [1,2,*], **Yiqun Xu** [1,2,*] **and Xiaoqing Luo** [4]

1    School of Marine Engineering, Jimei University, Xiamen 361021, China
2    Fujian Province Key Laboratory of Ship and Ocean Engineering, Xiamen 361021, China
3    Maritime College, Guangdong Ocean University, Zhanjiang 524055, China
4    College of Ocean and Meteorology, Guangdong Ocean University, Zhanjiang 524088, China
*    Correspondence: yu1202@hotmail.com (H.Y.); xnzh11189@163.com (Y.X.)

**Abstract:** Coastal governments have been preventing and controlling pollution in the marine environment by enhancing the construction of hardware and software facilities. The dispatch of offshore oil spill cleaning materials must be upgraded and optimized to cope with repeated offshore oil leak incidents while simultaneously improving cleaning efficiency and the ability to resist oil spill hazards. Accordingly, we set up a multiobjective optimization model with time window constraints to solve the scheduling optimization problem of offshore oil spill accidents with multiple locations and oil types. This model integrates the minimal sum of fixed costs, fuel consumption costs, maximum load violation costs, and time window penalty costs to solve the scheduling optimization problem of an offshore oil spill accident. An improved genetic algorithm is designed to solve the proposed mathematical model effectively and to make a scientific decontaminated decision-scheduling scheme. The practicality of the model and algorithm is validated by using a specific instance, demonstrating that the suggested method can effectively solve the schedule optimization problem for cleaning materials.

**Keywords:** emergency management; offshore oil spill; decontamination material scheduling; improved genetic algorithm; time window; multiobjective optimization



## 1. Introduction

In the exploration and development of offshore oil resources, oil spill accidents may be caused by maritime disasters, such as drilling blowouts, pipeline leakages, and ship collisions [1,2]. Owing to the particularity of the marine environment, oil spills have a remarkable negative effect on the ecosystems and economies of coastal areas. At present, offshore oil spill accidents are becoming more common at the domestic and international levels. For example, on 20 April 2010, the Deepwater Horizon blowout in the Gulf of Mexico of the United States led to a 200 km long and 100 km wide oil floating zone, which lasted for a long time and spread quickly [3]. On 16 July 2010, a large amount of crude oil leaked and caused a fire in the oil tanker area near Dalian Xingang, Liaoning Province, which caused severe damage to the marine ecological environment. On 11 June 2011, an oil spill in Bohai Bay resulted in coastal water pollution of approximately 840 km$^2$ [4]. On 23 November 2013, oil leakage and explosion occurred in an offshore oil pipeline in Qingdao, which resulted in residual oil entering the sea, and a large area of water around Jiaozhou Bay suffered serious oil pollution [5]. On 6 January 2018, Sanji and Changfeng Crystal collided about 296 km east of the Yangtze River estuary in Shanghai, which formed an oil slick of 10 km$^2$ [6]. On 27 April 2021, on the way from Port Sudan to Qingdao, the Panamanian general cargo ship Sea Justice collided with the Liberian tanker A Symphony, which was anchored in the waters southeast of Chaolian Island in Qingdao. Thus, the bow of the ship "Yihai" was damaged, and about 9400 tons of cargo oil leaked into the

sea, causing a particularly serious ship pollution accident, with the leakage of cargo oil worth about 22 million yuan [7]. In 2013, the incidence of Magnolia refinery spills and Rayong oil spills in the USA and Thailand spread about 680 and 43 tons of oil into the sea, respectively [8]. In 2012, the incident of Arthur Kill storage tank oil spills in the USA spread about 1090 tons of oil into the sea [8]. These oil spills have caused serious environmental pollution and ecological damage to offshore areas and have led to substantial loss of property and life to the enterprises and institutions involved. In recent years, the International Tanker Owners Pollution Federation Limited provided statistics showing that the number of oil spills was reduced because of the pollution control technology progress and tighter regulations introduced [9]. However, small oil spills continue to occur from time to time, causing incalculable damage to the environment [10]. Given the remarkable harm that oil leak accidents bring to marine ecosystems and coastal countries, making scientific and effective judgments about oil spill control is especially vital after an oil spill occurs. Therefore, emergency management in complex environments has become the focus of global attention.

The emergency response after the occurrence of offshore oil spills is crucial. The emergency resources on land must be coordinated to scientifically perform the subsequent cleanup. However, in reality, the decision maker often encounters unreasonable allocation of emergency resources. For example, a senior Chinese maritime safety official taking part in the emergency response to the ConocoPhillips oil spills admitted that, owing to the lack of theoretical decision support, the overall allocation of oil spill emergency resources depends on subjective experience, causing a slow and even chaotic emergency response. Serious oil spill emergency situations require the study of emergency resource scheduling to improve the efficiency of actual emergency responses [11]. A set of catastrophic oil spills is creating a global shockwave across industries, governments, and academia [12]. At present, the emergency management of oil spills has become a new research field. Researchers have conducted many studies in the areas of emergency management, risk and influence evaluation, and response technology growth [13–16]. However, research relevant to the study of offshore oil spill management and decision development is limited.

Research on emergency resource management for oil spills is still in its infancy. Offshore oil spill research typically focuses on cleanup measures, environmental impacts, materials developed to absorb oil, and support systems used to monitor and predict oil spills [17–21]. In addition, some researchers have realized good academic performance in the allocation of emergency resources in emergencies. Compared with the research on marine emergency resource allocation, land-based emergency resource allocation has made remarkable progress [22]. However, studies on land-based emergency resource scheduling concentrate on emergency medical treatment [23], facility positioning and route planning [24], and the allocation of emergency supplies [25].

Compared with land-based emergencies, maritime emergencies are highly maneuverable owing to complex weather and sea conditions, and the rescue target drifts over time or ultimately causes secondary accidents [26]. To reduce the influence of negative oil spills on the ecological environment, the study of offshore oil spill accidents involves the location storage of emergency supplies, harm to the human body, and recovery of oil spills. The logistics cost of emergency supplies is the main objective of oil spill rescue to avoid redundant expenses caused by excessive rescue points [27]. Considering maritime hazards and the specific circumstances of maritime rescue, Wang et al. [28] proposed a two-stage cooperative scheduling model for marine emergency resources. Huang et al. [29] presented a multiobjective optimization model for oil spill emergency resource scheduling on the basis of assumptions similar to those of Wang et al. Taking river chemical leakage as the research object, Liu et al. [30] established a framework for marine emergency supply allocation based on time-varying supply and demand constraints. This framework realizes the allocation of emergency supplies and the minimization of emergency response times. Hao et al. [31] used a triangle fuzzy approach to show the uncertainty of emergency resource demand and dispatch time during oil spill response. Garrett et al. [32] developed a mixed-integer



linear programming model to solve the dynamic resource allocation difficulties of the Arctic oil spill response network and provide emergency assurance for energy exploration. However, the previous study relied its emergency resource allocation model on the number of emergency supplies and scheduling time uncertainty, ignoring the complexity of marine oil spills, such as the time window, variety of oil spills, oil spill operation time, and the ideal time for oil spill recovery. It also ignores the characteristics of the entire cost of oil spill cleanup. This study found that the response to offshore oil spills is made after considering many elements. The type of oil (heavy crude oil, light crude oil, etc.) and the quantity of oil are the key elements influencing the oil spill clean-up process [33,34]. Light oil is volatile, flammable, and easy to clean up when oil spills occur, but heavy oil is not. This study takes the leakage of heavy crude oil as an example. Weathering due to wind, waves, and currents affects the recovery schedule of offshore oil spills. The emergency response to an offshore oil spill accident requires a rapid response and quick decision; otherwise, the area of oil spill diffusion and total cost of cleanup greatly increase [35–37]. Therefore, the decision-making problems of multisite and multi-type oil spills must be studied in depth. In this study, a multiobjective mathematical model for minimizing the total cost of offshore oil spill accidents integrated with the features of offshore oil spill accidents is established by considering uncertainties, such as the time window of oil spill clean-up operations, the demand for emergency supplies, and the scheduling time. An enhanced genetic algorithm is proposed to solve the multiobjective model, and an instance is given to analyze the effectiveness of the proposed approach.

The main contribution of this study is a multilocation and multi-oil-type scheduling optimization model for emergency supply in response to offshore minor oil spills (typical size of a dozen gallons) that considers overall dispatching costs and oil spill recovery time. This process can be achieved by performing the following steps.

1.  An optimization model for dispatching multilocation emergency supplies in response to small offshore oil spills that considers the total dispatching cost and oil spill recovery time is established.
2.  The interrelationships between the decision-making environment and the ground-breaking consideration of multisite cleanup of small oil spills are critical for an oil spill emergency response.
3.  Considering the timing of different types of oil spill recovery, this study uses the corresponding batch delivery time window to adjust the emergency operation and transportation of oil spill emergency supplementary resources.
4.  An improved genetic algorithm (IGA) is proposed to optimize the scheduling model of oil spills and sewage disposal materials in multiple offshore locations. This improves the efficiency and convergence speed of the calculation.

The rest of this paper is organized as follows. Section 2 presents the backdrop for the study problem. Section 3 establishes the mathematical model. Section 4 introduces the model's optimization approach. Section 5 discusses the case study and analysis. Section 6 summarizes the findings and suggests future research directions.

## 2. Background of the Problem

Marine oil spills refer to oil entering the marine environment, particularly crude oil and its related refined products [38]. Oil is defined as any form of petroleum, such as crude oil, fuel oil, sludge, residue, and refined commodities by the 1990 International Convention on Oil Pollution Preparedness, Response, and Co-operation. The spilled oil brings severe pollution (i.e., devastating adverse effects) to the marine ecology and damages the marine environment with the risk of fire. Timely and reasonable dispatching of oil spill emergency supplies is crucial to reduce potential safety hazards and damage to the marine ecology caused by oil spill accidents. In accordance with the comprehensive evaluation of oil spill volume, duration, speed, toxicity, and sensitive resources in the sea, offshore oil spill accidents can be divided into five grades: general accident, moderate accident, relatively severe accident, severe accident, and major accident [39,40]. This

research considers decision making in the event of an oil leak. Different sorts of oil spills, weather circumstances, marine conditions, and oil spill scales necessitate different materials, which include the following three types:

(1) Materials for blocking oil spills, mainly for the containment boom, are used for containment, oil spill diversion, and potential oil spill prevention.

(2) Materials for the recovery of oil spills, mainly for the oil collector, are used to recover water oil spills and oil and water mixtures.

(3) Other items mainly include adsorption materials and chemical and biological treatment agents, which are used to reduce damage caused by oil spills and accelerate the recovery of damaged waters.

When oil spills occur in the sea, a professional emergency rescue ship should be dispatched to perform the responsibilities of emergency command, containment, recovery, and storage of the oil spills. An oil spill cleaning vessel is a rescue ship that sets sail from the port and gets near the oil spill accident site to block, retrieve, and treat the oil leak. Oil spill response materials are transported by the ship. All the ships investigated in this research are oil spill cleanup ships. Oil spills in the sea are mobile owing to environmental factors, such as wind, waves, and currents. Therefore, multiple demand points for emergency supplies should be considered when formulating an emergency dispatch plan. When an oil spill accident occurs, the onshore oil spill emergency material reserves can provide emergency materials for the ship. The entire process of dispatching emergency supplies usually involves the oil spill cleanup ship loading the cleanup materials from the oil spill emergency supply storage, sailing to the oil spill accident site for the cleanup process, and returning to the dock after completing the task. A previous study showed that, under the influence of sea conditions, such as wind, wave, and flow, different degrees of weathering occur during oil spills, leading to remarkable changes in the physical and chemical properties of oil spills and affecting the treatment effect of oil spills. In this study, three types of offshore heavy crude oils were used as examples to explore the optimal disposal time window after offshore oil spills to provide decision support for offshore oil spill emergency treatment. Several experimental studies have shown that oil spills decrease rapidly with time in the first 6 h and then tend to be stable, which is contrary to the trend of oil spill water content changing with time [41]. When the viscosity of the three types of crude oil exceeds 10,000 MPa·s after 6 h, the oil spills adhere to the oil collection opening and cause blockage in the recovery process. Only a small number of oil spills are randomly brought into the oil collection head by water. Therefore, recovery efficiency is reduced and fluctuates greatly. In this study, three types of crude oils were selected as the oil information for offshore oil spills, which were labeled A, B, and C. We considered these crude oils because they are toxic to the environment. Their parameters are shown in Table 1. Previous research results show that the recovery rate of oil spills in offshore areas decreases sharply with changes in oil spill time. A is easy to recover, followed by B, while C is difficult to recover. The actual recovery efficiencies of A, B, and C after 6 h are only 12.2%, 3.5%, and 4.8%, respectively. Combined with the characteristic curve, the optimal disposal window for the recovery of A is 6 h; B, 5 h; C, 3 h. Therefore, A B, and C are used as the reference for the subsequent calculation examples of oil spill cleanup decisions [42].

**Table 1.** Parameter information of crude oil.

| Oil Samples | Density (20 °C)/(Kg·m$^{-3}$) | Viscosity (20 °C)/(MPa·s) |
| :---: | :---: | :---: |
| A | 933.4 | 437.7 |
| B | 926.9 | 581.2 |
| C | 934.0 | 1722.3 |

If frequent oil spill accidents occur in the offshore area, then maritime supervision departments need to conduct real-time supervision on the current marine danger situation

and formulate scientific and effective cleanup emergency plan to improve the ability of the sea to resist risks. Scientific and effective decision-making refers to the timely and efficient deployment of cleaning vessels and cleaning materials in accordance with the situation of random new accidents without affecting the emergency efficiency of the established scheme so as to satisfy the demands of the accident point with the highest quality to ensure the efficiency of the whole cleaning system. The dispatch of marine cleaning materials is determined by the specific emergency needs of the oil spill accident site. It is the transportation of a variety of emergency materials from the emergency base near the port to each accident site under limited transportation conditions. The time and quantity requirements of emergency materials for each accident site must be considered.

This study adopts a comprehensive planning method to transform the scheduling problem of oil spills and sewage materials into a multiprocess scheduling problem and makes efficient material scheduling decisions when small offshore oil spill events occur in multiple locations and oil types. The specific method is shown in Figure 1. We assume that the time experienced by all behavior in the entire scheduling process, from the time when the first accident occurs to the time when all emergency material needs of the last accident are met, is included in the same timeline. The material warehouse of the emergency base is located next to the wharf of the port area. The decontaminant materials are sufficient to satisfy the needs of many small- and medium-sized oil spill accidents. Oil spill decontaminant vessels are on standby beside the wharf, and the number of decontaminant ships is sufficient to satisfy the needs of many oil spill accidents in the sea area. The shore-based command center makes emergency material dispatching decisions in accordance with emergencies that have occurred within the last 6 h. This study focuses on the ship scheduling problem with the sum of the fixed cost, fuel consumption cost, maximum load violation, and penalty cost of the time window of the cleaning operation ship under the condition that multisite and multitype oil spill accidents in a certain water area can be effectively controlled within a period of emergency. Only one decontamination vessel should be used for decontamination at each accident site, and it should return to the distribution center after finishing the decontamination task. The time window of the accident point [$l$, $r$] is a time-limited area. The left time window denotes that the oil spill phenomenon will be delayed for a certain period after the occurrence of maritime emergencies, such as collisions and grounding [43,44]. The right time window refers to the optimal cleaning time, considering the recovery time of oil spills after the occurrence of an oil spill accident. The cleaning vessels should perform the cleaning operation within the time window required by each disaster site. No penalty cost is incurred if the cleaning vessel arrives early, but the cleaning operation can only begin after the time window has been set aside. The penalty cost is usually given to a breach of a set of rules, such as cleanup costs and harm to environmental assets and the local economy. However, it is absolutely not allowed to exceed the right time window, $r$. The corresponding penalty cost will be incurred once it exceeds the right time window, $r$.

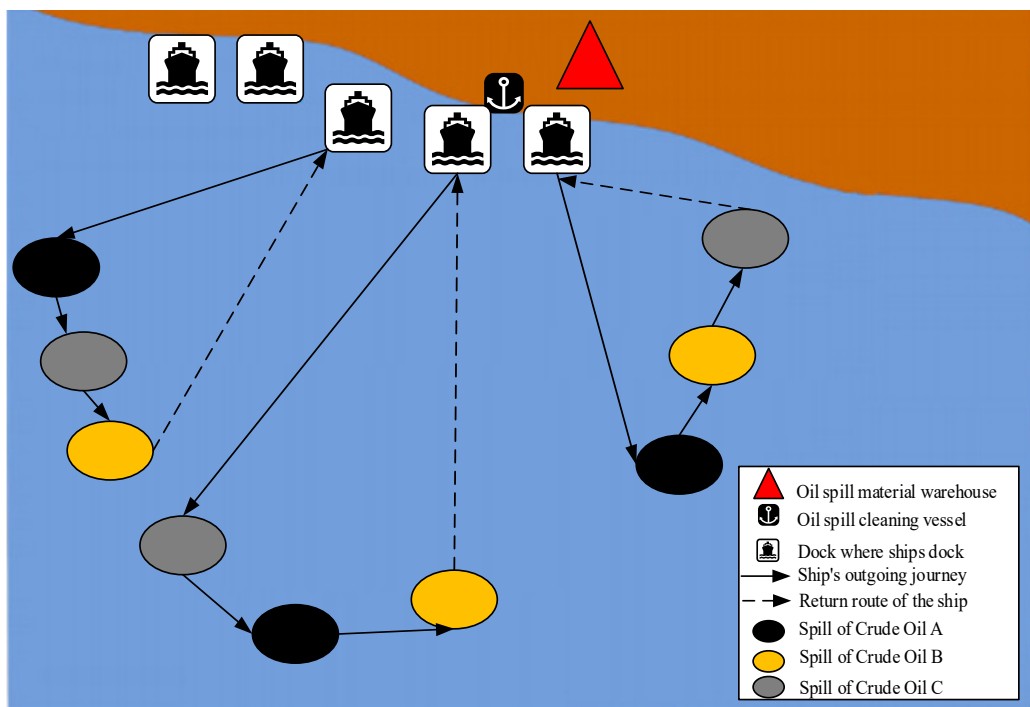

**Figure 1.** Schematic of offshore oil spills and sewage cleaning material scheduling.

## 3. Model Building

### 3.1. Model Assumptions

Considering the characteristics and operation procedures of oil spill accidents, the optimization of oil spill material scheduling requires multiple assumptions and constraints. The description and hypothesis of this study are as follows: In accordance with the characteristics of the research problem and the analysis of emergencies in the offshore area, the following model assumptions are made:

(1) In the case of general or small offshore oil spill accidents, the demand for emergency supplies is minimal, and the supply of shore-based points can meet the demand. In this case, emergency time is the most important factor. Various oil spill emergency materials, such as oil spill dispersants, oil booms, and oil absorption felt, are available in shore-based material storage. These materials are compatible in nature and can be loaded and transported together. In this study, the materials are packed and processed to form the decontamination-resource package when the demand for spilled oil materials is counted, and the quantity is measured in buckets during calculation. The total amount of shore-based storage completely meets the total demand.

(2) The type and quantity of emergency supplies needed at each accident point should be determined by the actual emergency situation known by the ship–shore communication system and the Geographical Information System.

(3) The loading and unloading times of materials account for a small proportion of the entire material scheduling process. The distance from the transport of materials from the emergency base to the cleaning vessel is extremely short, which has a limited impact on emergency efficiency. The cost and time of these factors are ignored to simplify the analysis.

(4) The vessels from the emergency center to the accident spot transport goods bear the roles of wind and waves. However, the entire process of emergency response and the environment do not change. The differences can remain stable despite the different carriers in different emergency bases, with accident points back and forth between the speed rates. A specific rate calculation method can be referenced.

(5) Territorial management is implemented between emergency bases. Loading between cleaning vessels is prohibited. The loading and unloading of supplies are not allowed

to change vessels. Only one oil spill recovery vessel is needed for each accident point to complete the cleaning task. Cooperative operation of multiple vessels is unnecessary.

(6) The loading capacity of the cleaning vessel and the demand for all types of oil spill materials can be arranged by a unified unit, and the materials at different accident points are forbidden from being mixed in the same transport ship.

(7) The loading capacity of the cleaning vessel is sufficient, and the sum of the demands of each customer on each distribution path does not exceed the cargo capacity of the ship. The needs of each site must be met, and only one cleanup vessel can perform one mission.

(8) All cleaning vessels must return to the dock for standby after completing cleaning tasks.

(9) All the functions constructed in the model are continuously differentiable convex functions. Under the condition of effectively controlling oil spill pollution and related constraints, the total dispatching cost of emergency oil spill materials, considering the time window problem, is minimized as the emergency target [45,46].

### 3.2. Associated Symbols and Definitions

$g_i$: Demand for emergency supplies at oil spill sites $i$;

$h_i$: Amount of oil recovered at the oil spill accident point $i$;

$d_{ij}$: Distance from the accident point $i$ to the accident point;

$t_{ij}$: Transport time of the cleaning vessel from the point $i$ of accident to the point of accident $j$;

$v$: Speed of cleaning vessel transport;

$u_1$: Fixed cost of dispatching a cleaning vessel;

$u_2$: Transport cost per kilometer of the cleaning vessel;

$t_i$: Time when the cleaning vessel arrives at the accident point $i$;

$l_i$: Left time window of oil spill accident point;

$r_i$: Right time window of the oil spill accident point;

$w_1$: Unit penalty cost of overloading a cleaning vessel;

$w_2$: Unit penalty cost for violation of the incident point right time window;

$z$: Total cost of oil spill cleanup;

$Q$: Maximum carrying capacity of a cleaning vessel.

Decision variables

$$x_{ijk} = \begin{cases} 1 \\ 0 \end{cases} \quad \text{The cleanup vessel } k \text{ sails from accident point } i \text{ to accident } j$$

$$y_{ik} = \begin{cases} 1 \\ 0 \end{cases} \quad \text{Materials from accident point } i \text{ to accident point } j \text{ are transported by cleaning vessel } k.$$

### 3.3. Establishment of Scheduling Model

The mathematical model for the transportation and distribution of oil spill cleanup materials in multiple locations and multiple oil types in the offshore area, considering time and cost factors, is expressed as follows:

$$z_1 = \sum_{i=1}^{N} \sum_{j=1}^{N} \sum_{k=1}^{M} u_1 x_{ijk} \tag{1}$$

Fixed cost of cleaning vessel dispatch

$$z_2 = \sum_{i=1}^{N} \sum_{j=1}^{N} \sum_{k=1}^{M} u_2 d_{ij} x_{ijk} \tag{2}$$

Fuel consumption costs incurred by cleaning vessels for transporting materials

$$z_3 = w_1 \sum_{k=1}^{M} \max \left[ \sum_{i=1}^{N} (g_i + h_i) y_{ik} - Q, 0 \right] \tag{3}$$

Penalty cost for breach of maximum carrying capacity of cleaning vessel

$$z_4 = w_2 \sum_{i=1}^{N} \max(t_i - r_i, 0) \tag{4}$$

Objective function

$$\min z = z_1 + z_2 + z_3 + z_4 \tag{5}$$

Constraints

$$\sum_{i=1}^{N} (g_i + h_i) y_{ik} \le Q, \tag{6}$$

$$\sum_{k=1}^{M} y_{ik} = 1, \tag{7}$$

$$\sum_{j=0}^{N} x_{ijk} = y_{ik}, \tag{8}$$

$$\sum_{i=0}^{N} x_{ijk} = y_{ik}, \tag{9}$$

$$\sum_{i=0}^{N} \sum_{k=0}^{M} y_{ik} = N, \tag{10}$$

$$\sum_{i=0}^{N} x_{iok} = \sum_{j=0}^{N} x_{ojk}, \tag{11}$$

$$t_i \le r_i, \tag{12}$$

$$t_j = \sum_{i=0}^{N} \sum_{k=1}^{M} (\max\{e_i, t_i\} + s_i + t_{ij}) \times x_{ijk}, \tag{13}$$

$$t_{ij} = \frac{d_{ij}}{v}, \tag{14}$$

$$x_{ijk}, y_{ijk} \in \{0, 1\}. \tag{15}$$

Formula (5) denotes the optimized objective function, indicating the sum of the dispatching cost, transportation fuel consumption cost, and the punishment cost of violating the maximum load, which indicates that the vessel's structure is overstressed, and the right time window of the oil spill accident point is the least.

Formulas (6)–(15) are the constraint conditions. Constraint (6) denotes that the quantity of cargo transported by each cleaning vessel does not surpass the maximum carrying capacity of the vessel. Constraint (7) indicates that only one decontamination vessel is required to complete the decontamination work at the accident point, and only one decontamination vessel can transport spilled oil materials. Constraints (8) and (9) indicate that each accident point can only be in one cleaning path. Constraint (10) denotes that all the materials at the accident point are completed by the cleaning vessel at the dock. Constraint (11) indicates that each cleaning vessel must return to the dock and stand by when cleaning is completed. Constraint (12) indicates that the time required to reach the accident point cannot exceed the time window of the accident point. Constraint (13) represents the time required to reach the accident point. Constraint (14) refers to the sailing time of the cleaning vessel

from the oil spill accident point to the oil spill accident point. Constraint (15) represents a 0–1 variable.

## 4. Research Methods

### *4.1. Genetic Algorithm (GA)*

GA can effectively solve scheduling problems. It is a random search algorithm based on natural biological selection and genetic mechanisms. Unlike the traditional search algorithm, GA optimization is an iterative process [47,48]. The algorithm maps the search space to the genetic space and maps the parameter variables to chromosomes. Every factor of the vector is called a gene, and all chromosomes constitute the population. The algorithm evaluates each chromosome based on the specified objective function and gives a suitable value based on the results. In this mechanism, the basic characteristics of individuals in each generation can be passed on to the next generation through chromosomes. In the next generation, the design schemes representing a population can be copied and crossbred with each other, and change with a certain probability. Hybridization tends to be undertaken by the best individuals in the population, and the offspring produced by the combination of the best characteristics of the matched individuals have better characteristics than the parent generation, resulting in better solutions. The optimization variables of classical GA are described by binary codes, which are connected together to form chromosomes. When an initial population is created, binary strings representing individuals are randomly generated within a certain word-length limit. The crossover operator acts on the two chromosomes selected in accordance with the crossover probability, randomly selects the crossover position, exchanges the binary values corresponding to these positions on the two chromosomes, and generates two new individuals. The mutation operator acts on individuals randomly chosen based on the mutation probability. The mutation bit is randomly selected, and the binary value of the bit is reversed to generate a new individual.

### *4.2. Improved Genetic Algorithm (IGA)*

GA transforms all types of engineering problems to be solved into coding models. All selection, crossover, and mutation are implemented for genes in individuals, which have nothing to do with the physical importance and characteristics of the original problem. This method has a wide range of adaptability and strong portability. GA has strong parallelism and can quickly search for an optimal solution in a large solution space, so deriving a global optimal solution is easy. However, the basic GA sometimes has problems, such as premature convergence, slow evolution speed, finding suboptimal solutions, and falling into the local optimum, so it needs to be improved. GA have obvious advantages compared with traditional algorithms. It does not require function continuity, nor does it require the function to be derivable. GA has the characteristics of simple implementation and low requirements for objective function. It is a global population search algorithm that simulates biological evolution and uses a natural evolution mechanism to represent complex phenomena. However, GA has some problems; it may exhibit a "premature" phenomenon, and the convergence speed is sometimes relatively slow. This study optimizes certain genetic operators.

### 4.2.1. Chromosome Coding

Prüfer sequence is used to encode the chromosomes of the IGA. In accordance with the principle that the minimum spanning tree with degree constraint has a number limit on the connected edges of nodes, the degree limit should be considered when using the generated Prüfer sequence as the initial population to avoid the generation of infeasible solutions and to increase the efficiency of the solution. Suppose the degree constraint sequence of a problem is $b = (b_1, b_2, \cdots, b_n)$, then a sequence of numbers can be obtained in accordance with the degree constraint of each node $s = \{1, 1 \cdots 1, 2, 2 \cdots 2, 3, 3 \cdots 3, i, i, \cdots i, n, n \cdots n\}$. The number of the number $i$ in the sequence is $b_i - 1$. The specific operations to obtain the initial population of Prüfer sequence are as follows: Randomly take $n - 2$ numbers from

the sequence *s* to form a new sequence. Any random permutation combination containing a $n - 2$ number arrangement obtained from the above sequence can be used as the Prüfer number corresponding to a spanning tree and meet the degree constraint requirements of each node. The operation of picking a random $n - 2$ number from the sequence *s* is repeated until the maximum number of individuals required to initialize the population $N_p$ is reached. In this way, the initial population is obtained, fully satisfying the degree constraint of each node, and does not generate infeasible solutions.

### 4.2.2. Fitness Function

The advantages and disadvantages of individual populations are evaluated and distinguished using the fitness function of GA, and the fitness value is the basis of genetic selection. The objective function of the model is composed of the fixed cost dispatched by the cleaning vessel, the transportation cost of the cleaning materials, the penalty cost of the violation of the maximum load of the cleaning vessel, and the right time window of the accident point. Combined with the requirements for oil spill clean-up, the smaller the objective function, the better the result. The fitness function is defined as the reciprocal of the objective function to retain the good result in the selection operation, which is expressed as $f(x) = 1/f$.

### 4.2.3. Genetic Operators

(1)　Choice

The selection process of the IGA adopts a relatively fair roulette strategy. The basic idea of roulette selection is that the probability of each individual being selected is proportional to its fitness. If the traditional roulette selection strategy is adopted, then individuals with extremely good fitness in the previous generation's population may be discarded. Therefore, this study adopts the roulette strategy plus the selection method of retaining optimal individuals. Specifically, the optimal individual is retained first, and then the next generation of individuals is selected in accordance with the roulette wheel between the parent and child generations.

(2)　Crossover and mutation operators

The selection of crossover possibility $p_c$ and mutation possibility $p_m$ among the coefficients of GA is the core to affect the action and performance of the algorithm, which influences its convergence directly. The greater the probability of crossover, the faster new individuals are produced. In the case of an extremely large crossover probability, the probability of the genetic pattern being destroyed is greater, so the individual structure with high fitness will soon be damaged. However, if it is extremely small, the search process will be slow and even stagnate.

For an extremely small mutation probability, a new individual structure is difficult to generate. In the case of an extremely large value, the GA becomes a purely random search algorithm. When the fitness of every individual in the population converges or tends to be the local optimum, we increase the two probabilities. When population fitness is more dispersed, we reduce it. If the fitness is higher than the population average fitness of the individual, with a lower possibility of crossover and mutation, then the excellent performance of the individual is carried into the next generation. For below-the-average fitness of the individual, high crossover and mutation probabilities are used to eliminate individuals with poor performance. Therefore, the adaptive $p_c$ and $p_m$ can provide the best sum relative to a solution. An adaptive GA can maintain the diversity of the population and ensure the convergence of GA. As the parameters of each genetic operator are improved, the algorithm can adapt to the characteristics of each stage of population evolution, and the optimization efficiency and quality of the algorithm are improved.

The crossover probability is calculated as follows:

$$p_c = \begin{cases} p_a - \frac{(p_a - p_b)(f' - f_{mean})}{f_{max} - f_{mean}} & f' \geq f_{mean} \\ p_a & f' < f_{mean} \end{cases},$$ (16)

where $p_a = 0.8$, and $p_b = 0.5$. $f_{max}, f_{mean}$ represent the maximum fitness and average fitness of individuals in the current population, respectively. $f'$ represents the greater fitness of the two selected individuals.

The calculation formula for mutation probability is as follows:

$$p_m = \begin{cases} p_u - \frac{(p_u - p_v)(f_{max} - f)}{f_{max} - f_{mean}} & f' \geq f_{mean} \\ p_u & f' < f_{mean} \end{cases},$$ (17)

where $p_u = 0.1$, $p_v = 0.002$, and the selection of mutation points is random.

*4.3. Algorithm Step*

The specific process of the IGA is shown in Figure 2, which is divided into 6 steps.

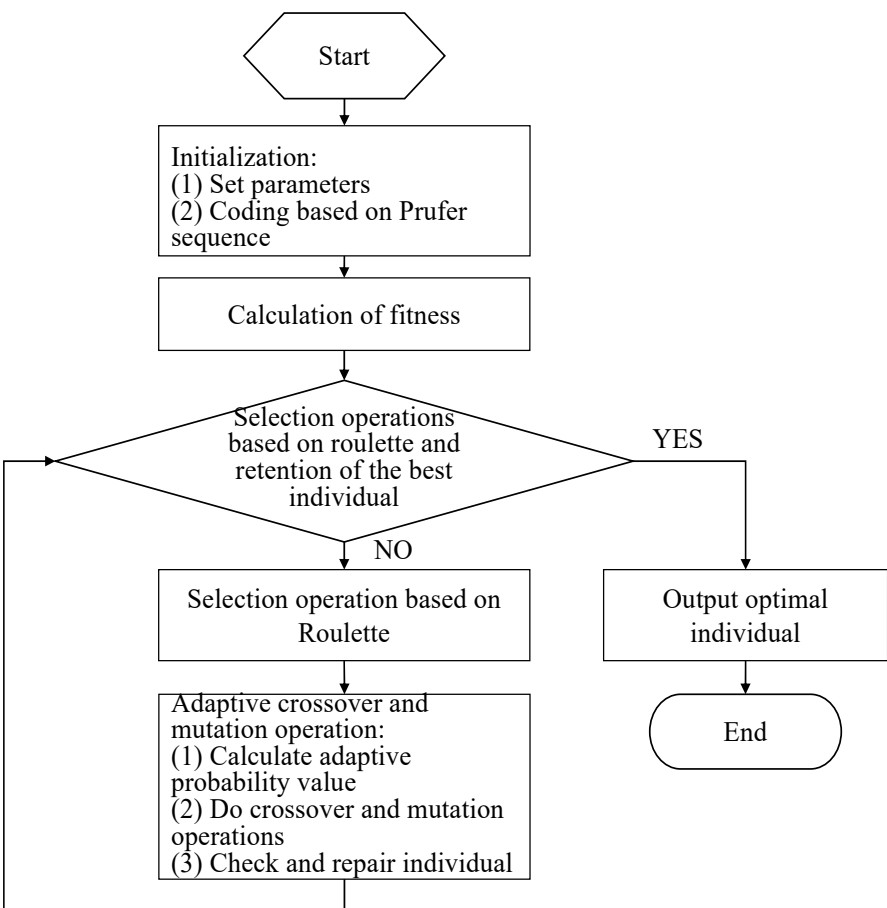

**Figure 2.** Flowchart of the IGA.

Step 1 uses binary encoding to generate the initial population, and $p$ individuals $\{X_i\}$ ($i = 1, 2 \ldots P$) are generated in accordance with Prüfer sequence encoding in the domain of function definition. The main coefficients of the algorithm are set.

Step 2 calculates the function value of each individual, the average value of the population function, and the fitness of each individual in the current population based on the fitness function set in this study.

Step 3 is the optimal preservation strategy. It first calculates the function value of each individual and then sorts it to find the optimal solution and the worst solution. If the function value of the optimal solution of the previous generation is larger than that of the current optimal solution, then the optimal solution of the previous generation is replaced by the current optimal solution. If the function value of the optimal solution of the previous generation is small, then the optimal solution of the past generation replaces the worst solution of the current generation.

Step 4 adopts the proportional selection method in accordance with the fitness of each individual. Individuals with small original function values in the early stages of evolution will have a greater probability of being selected through transformation, thereby maintaining the diversity of the population.

Step 5 calculates the crossover and mutation probabilities, selects two parent individuals for crossover in accordance with fitness, and selects the best two individuals in the current population to be inherited into the offspring.

Step 6 stops when the number of iterations is satisfied; otherwise, it adds 1 to algebra and proceeds to step 2.

## 5. Implementation of Simulation Experiment and Analysis

This section designs a simulation experiment of offshore oil spill cleanup scheduling with multiple locations and oil types to test and verify the practicality of the established model [49,50]. The proposed IGA can be used for the emergency decision making of small offshore oil spill accidents in complicated environments to decrease disaster losses.

### 5.1. Example Description

An oil spill material emergency base (numbered 0) is stationed in a bay that has sufficient storage of oil spill cleaning materials. A large number of vessels are passing in this area, and water operations are busy, so the oil spill risk is high, indicating that the damaging capacity is extremely destructive. Accordingly, the port area in the region is equipped with 10 emergency cleaning vessels. One day at 4 o'clock in the morning, 12 ship emergencies were found in the process of marine environment inspection in the offshore waters. After 1.5 h, 12 locations (numbered 1, 2 . . . 12) caused small oil spills in succession. The specific time of the oil spill is shown in Table 1. The 10 cleaning vessels berthed in the port area are all on standby, which requires the maritime supervision department to take scientific cleaning measures to prevent the pollution situation from expanding. The location of the emergency station or headquarters and the oil spill accident point, the demand of the affected point, the type of oil spills, the time window of the oil spill accident point, and the time of cleaning operation are obtained. In this study, the longitude and latitude coordinates of the oil spill accident points are generated on the map to show the simulation outcomes more intuitively. The coordinates of the oil spill accident points are processed as follows: only three digits after the decimal point of the longitude and latitude are kept. The calculation example requires the maritime regulatory department to formulate a scientific decontamination-related scheme and determine the best decontamination-related ship scheduling scheme to deal with all oil spill accidents in a short time and reduce decontamination-related costs as much as possible under the constraints of ship capacity and time window. The specific data related to the calculation example are shown in Table 2. The latitude and longitude of each vessel are processed to display the calculation results intuitively as follows: only the 3 digits after the decimal point of the longitude are retained to generate the abscissa coordinates, and only the 3 digits after the decimal point of the latitude are retained. The coordinates of each point after setting in accordance with this principle are shown in Table 1. The parameter settings in the IGA are given in Table 3.

**Table 2.** Data for the calculation example of oil spills.

| Serial Number | X Coordi-nates | Y Coordi-nates | Required Materials /Drum | Amount of Dirty Oil Per Barrel | Occurrence Time of Oil Spills (AM) | Right Time Window (AM) | Cleaning Operation Time/Min | Oil Spill Species |
|---|---|---|---|---|---|---|---|---|
| 0 | 100 | 0 | 0 | 0 | 4:00 | 10:00 | 0 | 0 |
| 1 | 1 | 40 | 18 | 12 | 4:24 | 8:24 | 60 | A |
| 2 | 4 | 60 | 18 | 16 | 4:16 | 8:54 | 80 | B |
| 3 | 6 | 110 | 25 | 10 | 5:23 | 7:23 | 60 | C |
| 4 | 50 | 130 | 19 | 20 | 5:18 | 8:38 | 100 | A |
| 5 | 70 | 30 | 22 | 13 | 4:19 | 9:19 | 60 | B |
| 6 | 90 | 5 | 19 | 10 | 5:01 | 6:41 | 80 | C |
| 7 | 120 | 14 | 18 | 12 | 4:20 | 8:20 | 60 | A |
| 8 | 150 | 35 | 16 | 14 | 5:21 | 10:01 | 80 | B |
| 9 | 160 | 190 | 14 | 7 | 4:15 | 6:15 | 60 | C |
| 10 | 127 | 170 | 19 | 10 | 5:24 | 9:36 | 48 | A |
| 11 | 140 | 149 | 24 | 11 | 4:15 | 9:35 | 40 | B |
| 12 | 63 | 157 | 20 | 8 | 5:22 | 7:52 | 30 | C |

**Table 3.** Parameter settings of the IGA.

| Serial Number | Parameter | Value |
|---|---|---|
| 1 | Population size | 200 |
| 2 | Evolution algebra | 100 |
| 3 | Crossover probability | $P_a = 0.8$ $P_b = 0.5$ |
| 4 | Mutation probability | $P_u = 0.1$ $P_v = 0.002$ |
| 5 | Generation gap | 0.9 |
| 6 | Number of cleaning vessels on standby/vessel | 10 |
| 7 | Maximum carrying capacity of a cleaning vessel/barrel | 100 |
| 8 | Speed of the cleaning vessel/km/h | 50 |
| 9 | Use cost of cleaning vessel/10,000 yuan | 100 |
| 10 | Transport cost per unit distance of cleaning vessel/yuan·Km$^{-1}$ | 70 |
| 11 | Penalty cost for breach of loading capacity (10,000 yuan·barrel$^{-1}$) | 5 |
| 12 | Penalties for violating the time window constraints (10,000 yuan·min$^{-1}$) | 1 |

*5.2. Experimental Results and Discussion*

The IGA and the mathematical model established above are combined to solve this example, and the IGA program is written in accordance with the algorithm flow shown in Figure 3. The solution results are described below. The optimal scheduling scheme and route optimization scheme of spilled oil materials are obtained after repeated debugging and operation. As shown in Figure 3, the number of cleaning vessels in use is 4, which are numbered as 1, 2, 3, and 4. The distribution path of the NO.1 cleaning vessel is 0 ->1 -> 2 -> 5 -> 0. The distribution path of the NO.2 cleaning vessel is 0 ->9 -> 10 -> 11 -> 8 -> 0. The distribution path of the NO.3 cleaning vessel is 0 ->3 -> 12 -> 4 -> 0. The distribution path of the no. 4 cleaning vessel is 0 ->6 -> 7 -> 0. The total cleaning cost of the dispatching scheme is calculated to be 11,942,653 yuan, in which the vehicle dispatch cost is 4 million yuan, the fuel consumption cost is 7,942,653 yuan, the

penalty cost of violating ship loading capacity is 0, and the penalty cost of violating the time window is 0.

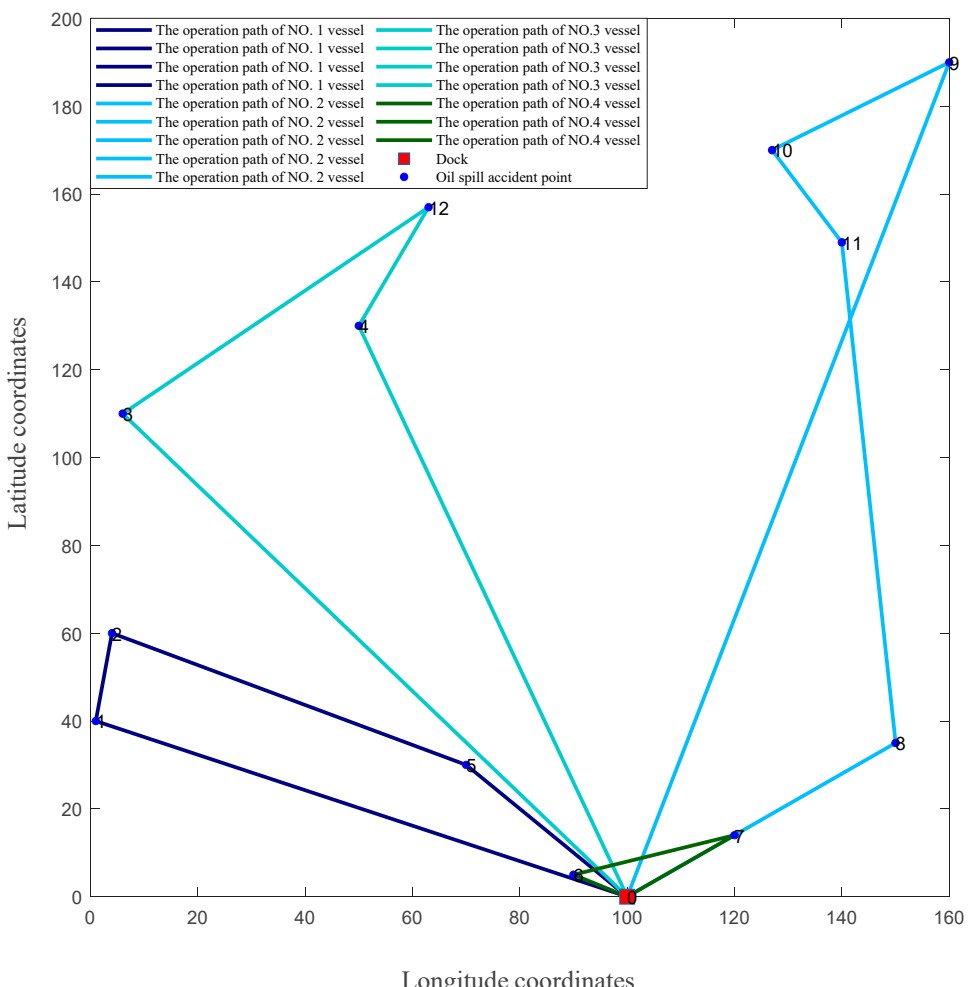

**Figure 3.** Scheduling scheme for waste cleaning materials based on IGA.

Standard GA and simulated annealing (SA) algorithms are selected in this study to compare the merits and demerits of the sewage disposal scheduling scheme. SA is a random optimization algorithm based on the Monte Carlo iterative solution measure [51,52]. It provides an effective approximate solution algorithm for multidimensional complex problems and overcomes the feature that other algorithms easily fall into a local optimal solution. The parameter settings of the standard GA and SA are shown in Table 4, and the parameters of the conditional constraints are given in Table 1. The optimization results of the three algorithms are obtained many times after debugging, as shown in Table 5. When the population size and iteration times are the same, the scheduling schemes calculated by the three algorithms all use four cleaning vessels. The penalty cost for violating ship loading capacity and time window is zero, and the scheduling schemes for cleaning vessels are slightly different. From the total cost of cleaning operations and fuel consumption cost of ship navigation, the optimal fitness values obtained by the IGA are obviously better than those of the standard GA and SA, whereas those of the SA and standard GA are similar.

**Table 4.** Parameter settings of GA and SA.

| Algorithm | Serial Number | Parameter | Value |
|---|---|---|---|
| GA | 1 | Population size | 200 |
| | 2 | Evolution algebra | 100 |
| | 3 | Crossover probability | $P_C = 0.9$ |
| | 4 | Mutation probability | $P_m = 0.05$ |
| | 5 | The generation gap | 0.9 |
| SA | 1 | Initial temperature | 3000 |
| | 2 | Final temperature | 0.01 |
| | 3 | Temperature attenuation factor | 0.98 |
| | 4 | Markov chain length | 100 |
| | 5 | Tolerance | 1 |
| | 6 | Step length factor | 0.3 |
| | 7 | A metropolis procedure always accepts points | 0 |

**Table 5.** Comparison of the optimization results of different algorithms.

| Algorithm | Number of Cleaning Vessels Used | Total Cost of Clean-Up/Ten Thousand Yuan | Fuel Consumption /Ten Thousand Yuan | Penalty Cost for Breach of Loading Capacity/Ten Thousand Yuan | Penalty Costs for Time Window Violations/Ten Thousand Yuan | Scheduling Scheme for Cleaning Vessel Operation |
|---|---|---|---|---|---|---|
| IGA | 4 | 11,942,653 | 7,942,653 | 0 | 0 | The operation path of the NO. 1 cleaning vessel is: 0 ->1 -> 2 -> 5 -> 0 The operation path of the NO. 2 cleaning vessel is: 0 ->9 -> 10 -> 11 -> 8 -> 0 The operation path of the NO. 3 cleaning vessel is: 0 ->3 -> 12 -> 4 -> 0 The operation path of the NO. 4 cleaning vessel is: 0 ->6 -> 7 -> 0 |
| GA | 4 | 12,287,934 | 8,287,934 | 0 | 0 | The operation path of the NO. 1 cleaning vessel is: 0 ->1 -> 3 -> 2 -> 0 The operation path of the NO. 2 cleaning vessel is: 0 ->5 -> 12 -> 4 -> 0 The operation path of the NO. 3 cleaning vessel is: 0 ->6 -> 7 -> 0 The operation path of the NO. 4 cleaning vessel is: 0 ->9 -> 10 -> 11 -> 8 -> 0 |
| SA | 4 | 12,263,579 | 8,263,579 | 0 | 0 | The operation path of the NO. 1 cleaning vessel is: 0 ->3 -> 12 -> 4 -> 0 The operation path of the NO. 2 cleaning vessel is: 0 ->11 -> 9 -> 10 -> 0 The operation path of the NO. 3 cleaning vessel is: 0 ->6 -> 7 -> 8 -> 0 The operation path of the NO. 4 cleaning vessel is: 0 ->5 -> 2 -> 1 -> 0 |

Figure 4 shows the convergence of the three types of curve. The IGA with an adaptive crossover mutation strategy greatly increases the offspring's population diversity and accelerates the convergence speed of the GA. The number of iterations is about 40 times optimal. The IGA shows strong optimization ability and convergence. The GA reaches

the optimal solution after about 65 generations of evolution. The number of iterations of the SA is about the same as that of the IGA, but the operation time is about five times that of the IGA, and its optimal solution is larger than that of the IGA. The three algorithms were independently run 100 times with different iterations to compare the performance and record the average value, standard deviation, and calculation time. This process was performed to further study the optimization performance of the IGA. The results are provided in Table 6. The IGA has advantages in terms of optimization results and optimization speed. The proposed IGA has certain advantages in optimizing the scheduling of oil spill materials.

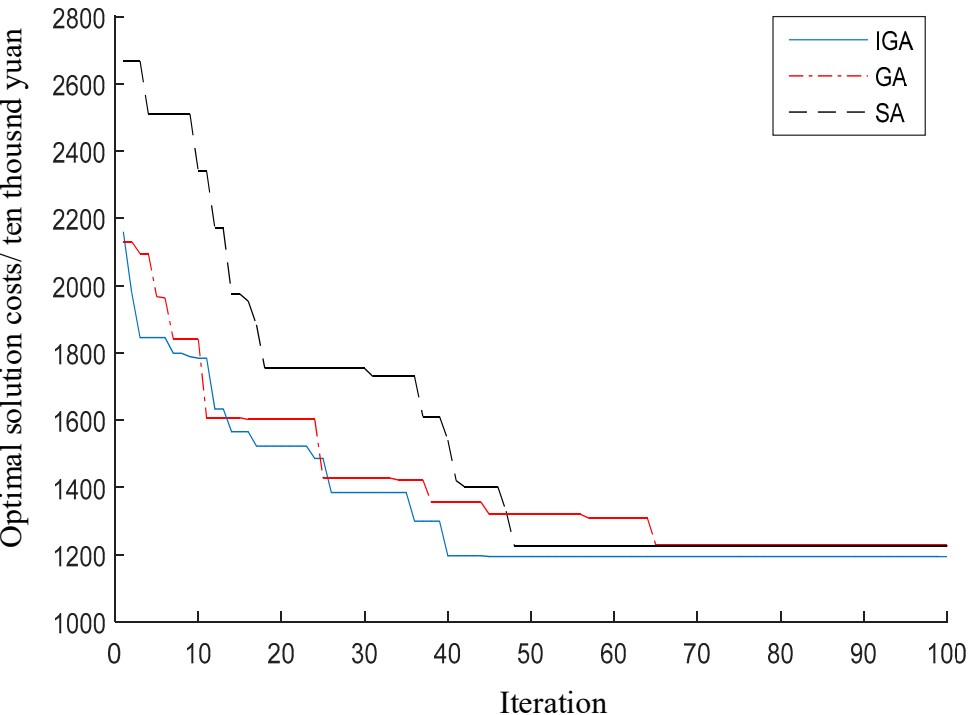

**Figure 4.** Iterative curves of the three optimization algorithms.

**Table 6.** Statistical results of the three algorithms running independently 100 times.

| Algorithm | Average Value/Ten Thousand Yuan | Standard Deviation | Operation Time/s |
| --- | --- | --- | --- |
| GA | 12,301,256 | 19.33456 | 287.3153 |
| SA | 12,288,968 | 16.4226 | 1202.8636 |
| IGA | 11,902,352 | 5.3426 | 221.2729 |

## 6. Conclusions and Future Research

The emergency management of offshore oil spill accidents is a complicated decision-making problem. How to deal with the complex environment to make scientific and effective decisions about oil spill cleanup and to decrease the loss of oil spills to the maximum extent has become a challenging problem faced by coastal countries. An optimization model of cleaning material scheduling for multilocation and multioil spills combined with the characteristics of the scheduling workflow of cleaning materials for oil spill accidents of small offshore vessels was established to address the problem of how to resist the risk of oil spills in offshore areas. An IGA was designed and validated for scheduling optimization to meet the demand of the oil spill accident point under the premise of giving attention to two or more things during cleanup, including fixed costs, transportation costs, violation of the maximum load, and the time window of the optimization target, to minimize the sum

of the penalty cost. SA, GA, and IGA were performed on real-life examples with the help of MATLAB software to reach the following conclusions.

(1) In comparison to the SA and regular GA, the IGA produces superior results and arrives at the best answer faster in the evolutionary process.

(2) The constructed multisite and multioil-type scheduling optimization model of oil spills and decontamination-related materials has universality. The designed hybrid GA has a high timeliness in solving the model, which can provide a scientific decision-making basis for solving small offshore multisite oil spill accidents.

This research is helpful in optimizing the linkage dispatching of emergency supplies when multiple offshore oil spill accidents occur simultaneously, improving the timeliness of emergency response, and reducing disaster losses. However, the material demand of the oil spill disaster spot changes dynamically due to the impact of the type of emergency. Accordingly, the scheduling model used in this study still needs to be improved in some areas, such as considering the diffusion and movement of stationary targets after the occurrence of actual oil spill accident points, allocating in accordance with emergency response requirements, and dealing with the material scheduling of large oil spill accidents, in a follow-up study. Considering complicated conditions when deploying maritime oil spill cleanup materials is an area of future research.

**Author Contributions:** Conceptualization, K.L.; Data curation K.L.; Methodology, H.Y. and Y.X.; Project administration, X.L. and K.L.; Software, K.L.; Supervision, K.L.; Writing—review and editing, K.L. All authors have read and agreed to the published version of the manuscript.

**Funding:** This study was supported by Fujian Provincial Science and Technology Planning Project (2020H0018; 2021H0020) and the Zhanjiang City Science and Technology Development Special Fund Competitive Allocation Project (NO.2021A05034).

**Institutional Review Board Statement:** Not applicable.

**Informed Consent Statement:** Not applicable.

**Data Availability Statement:** All data are provided in the article.

**Conflicts of Interest:** The authors declare no conflict of interest.

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
