# Peer review of "Scheduling Optimization of Offshore Oil Spill Cleaning Materials Considering Multiple Accident Sites and Multiple Oil Types"

_sustainability, doi:10.3390/su141610047_

Round 1

Reviewer 1 Report

This research is about fleet management for optimizing the cleanup of multiple small spills. I believe the research does not provide a major novelty but may attract the readership of the Sustainability Journal. The manuscript writing is poor and some sentences are difficult to understand.

11)      As some spills at early stages of occurrence may be more important than the older spills for cleanup, do you consider the oil weathering in your optimization algorithm?

22)      As the oil spills moves with wind and current, how do you take into account the real-time location of multiple oil spill in optimizing the cleanup operation?

33)      Please elaborate in manuscript on definition of penalty costs.

44)      The introduction is not organized and need to be rewritten.

55)      Line 32: you provide example of very large oil spills while your method is appropriate for multiple small spills. Please revise.

66)      Line 114: this statement may not be compatible with the global trend of fossil fuel consumption: "With the development of offshore water transportation and offshore oil development industry, the probability of offshore oil spill accidents occurring at multiple sites and with multiple types of oil gradually increases".

77)      Line 128: the numbered bullet are not related to introduction.

88)      Line 146: this paragraph should be omitted: "The Materials and Methods should be illustrated with adequate details …"

99)      Line 192: please provide references for: "Several experimental studies show that oil spill decreases rapidly …"

110)   Line 200: please explain in test that why these three oils are selected for this study.

111)   Line 221: a paragraph does not usually start with "Therefore".

112)   Table 1: Please specify the parameters of oil type C.

113)   Line 246: this sentence is vague: "but the cleaning operation can start only after the left time window l".

114)   Line 360: Please define the "violating the maximum load" in the text.

115)   Figure 3: please define in the legend, different colors of routes.

116)   Figure 4: please change the "optimal solution" axis to optimal solution costs and specify the unit.

117)   Line 621: please explain the "However, in reality, the material demand of oil spill accident point changes dynamically owing to the impact of the type of emergency".

118)   Please recheck the entire text for punctuation, grammar and technical term selection.

Author Response

11)      As some spills at early stages of occurrence may be more important than the older spills for cleanup, do you consider the oil weathering in your optimization algorithm?

Response:Thank you for your good observation. The optimization problem in our study considers oil weathering. After the occurrence of oil spill, with the passage of time, the density, viscosity and water content of oil products rise rapidly under the influence of volatilization, emulsification and air dissolution, thus affecting the work efficiency of oil spill cleaning. In this study, the optimal disposal window for recovery of crude oil A is defined as 6h, crude oil B as 5h, and crude oil C as 3h.

22)      As the oil spills moves with wind and current, how do you take into account the real-time location of multiple oil spill in optimizing the cleanup operation?

Response: Thank you for your good observation. We agree with you that wind speed and current play a vital role in spreading of oil spill. This is a big issue to locate the actual location of oil spill. For this reason, we have mentioned the specific sampling time. We will try to integrate this issue in model in our next follow-up study.

33)      Please elaborate in manuscript on definition of penalty costs.

Response:Thank you for your suggestion. Revised as suggested.

44)      The introduction is not organized and need to be rewritten.

Response:Thank you for your suggestion. Revised thoroughly.

55)      Line 32: you provide example of very large oil spills while your method is appropriate for multiple small spills. Please revise.

Response:Incorporated as desired along with large oil spills.

66)      Line 114: this statement may not be compatible with the global trend of fossil fuel consumption: "With the development of offshore water transportation and offshore oil development industry, the probability of offshore oil spill accidents occurring at multiple sites and with multiple types of oil gradually increases".

Response: Thank you for your suggestion. Removed from the manuscript.

77)      Line 128: the numbered bullet are not related to introduction.

Response: Thank you for your suggestion. Revised accordingly.

88)      Line 146: this paragraph should be omitted: "The Materials and Methods should be illustrated with adequate details …"

Response: Thank you for your good observation. Deleted as desired.

99)      Line 192: please provide references for: "Several experimental studies show that oil spill decreases rapidly …"

Response: Thank you for your good observation. Revision done.

110)   Line 200: please explain in test that why these three oils are selected for this study.

Response: Thank you for your good observation.We have selected oil spill types A, B & C because all these three are toxic to the environment. This issue is mentioned in the text accordingly.

111)   Line 221: a paragraph does not usually start with "Therefore".

Response: Thank you for your suggestion. Revised accordingly.

112)   Table 1: Please specify the parameters of oil type C.

Response:Thanks, modification done.

113)   Line 246: this sentence is vague: "but the cleaning operation can start only after the left time window l".

Response:Modified as desired.

114)   Line 360: Please define the "violating the maximum load" in the text.

Response: Revised as desired.

115)   Figure 3: please define in the legend, different colors of routes.

Response:Thank you for your suggestion. Modified as desired.

116)   Figure 4: please change the "optimal solution" axis to optimal solution costs and specify the unit.

Response: Thank you for your suggestion. Modified as desired.

117)   Line 621: please explain the "However, in reality, the material demand of oil spill accident point changes dynamically owing to the impact of the type of emergency".

Response: Sorry for the misunderstanding. Here we are trying to explain the fact that depending the oil spill types (severity) the requirement of material demand varied.

118)   Please recheck the entire text for punctuation, grammar and technical term selection.

Response: Thank you for your suggestion. Revised thoroughly.

Reviewer 2 Report

Dear Authors,
The article tries to optimize the scheduling of the oil spill cleaning procedures by the mathematical model using the IGA algorithm. The topic is interesting and promising, although the English language and scientific soundness are weak. Please find below a few suggestions to improve the article.
line 5 and 11: correspondence author?
line 13,14: unclear
line 27 the aim is missing
line 29: rephrase "marine emergencies"
32: home??
32: the year 2010 is not "nowadays"
56,57: is this a fact or guessing? reference?
61-65: reference
all text: I suggest following the journal's template regarding references and citations.
101: which study
105: unclear
107: which study?
129: explain and define "small"
146-152: ???
157: define "severe"
163: is this the aim? If so, it goes to the beginning of the introduction.
174-179: consider scientific soundness
192: reference of experimental studies missing!
200: where in table 1 is parameter C?
244: what are the odds to have 9 oil spills at the same time?
229: define
260: explain
355-357: ??
525: define "high"
528: is this guessing or prediction - on what base?
what is the cause?
534: obtained from where?
535: coordinates obtained from where?
table 3 and section 5.2: Suggesting calculating yuan to USD or perhaps Eur.
578: unclear
611: the conclusion needs improvement. 
- Perhaps the mentioning/comparison with the "Wärtsilä Oil Spill Response simulator" would be interesting to the reader.
Brgds.

Author Response

The article tries to optimize the scheduling of the oil spill cleaning procedures by the mathematical model using the IGA algorithm. The topic is interesting and promising, although the English language and scientific soundness are weak. Please find below a few suggestions to improve the article.

Response:Thank you for your kind recommendation.

line 5 and 11: correspondence author?

Response:The corresponding author of lines 5 and 11 of the article has been adjusted to maintain consistency

line 13,14: unclear

Response:Thank you for your good observation. Revised accordingly.

line 27 the aim is missing

Response:Thank you for your good observation. Revised thoroughly.

Line 29: rephrase “marine emergencies”

Response:Done as suggested.

32: home??

Response:Thank you for your good observation. Revised accordingly.

32: the year 2010 is not "nowadays"

Response:Thank you for your good observation. Modified as suggested.

56,57: is this a fact or guessing? Reference?

Response:Incorporated proper references.

61-65: reference

Response:Thank you for your good observation. Done.

all text: I suggest following the journal's template regarding references and citations.

Response:Thank you for your good observation. Done.

101: which study

Response:Mentioned now clearly.

105: unclear

Response:Thank you for your good observation. Revised.

107: which study?

Response:Mentioned now clearly.

129: explain and define “small”

Response:Explained now clearly.

146-152: ???

Response:Thank you for your good observation. Revised.

157: define “severe”

Response:Define as required.

163: is this the aim? If so, it goes to the beginning of the introduction.

Response:We have revised the part.

174-179: consider scientific soundness

Response:Revised accordingly.

192: reference of experimental studies missing!

Response:Thank you for your good observation. Revised.

200: where in table 1 is parameter C?

Response:C represents oil spill types,as showed in Table 1.

244: what are the odds to have 9 oil spills at the same time?

Response:Sorry for the misunderstandings. They are happened at different times; we have mentioned it in the Table 2.

229: define

Response:Thank you for your good observation. Revised.

260: explain

Response:Thank you for your good observation.Explained clearly now.

355-357: ??

Response:Thank you for your good observation.Revised.

525: define “high”

Response:Thank you for your good observation.Defined in the text as desired.

528: is this guessing or prediction - on what base? what is the cause?

Response:Emergency base means emergency base station or headquarters.
